# Solid Dispersion of Resveratrol Supported on Magnesium DiHydroxide (Resv@MDH) Microparticles Improves Oral Bioavailability

**DOI:** 10.3390/nu10121925

**Published:** 2018-12-05

**Authors:** Roberto Spogli, Maria Bastianini, Francesco Ragonese, Rossana Giulietta Iannitti, Lorenzo Monarca, Federica Bastioli, Irina Nakashidze, Gabriele Brecchia, Laura Menchetti, Michela Codini, Cataldo Arcuri, Loretta Mancinelli, Bernard Fioretti

**Affiliations:** 1Prolabin & Tefarm, Spin-Off Un. of University of Perugia, Via Dell’Acciaio 9, Ponte Felcino, 06134 Perugia, Italy; roberto.spogli@prolabintefarm.com (R.S.); maria.bastianini@prolabintefarm.com (M.B.); 2Department of Chemistry, Biology and Biotechnologies, University of Perugia, Via Elce di Sotto 8, 06123 Perugia, Italy; francesco.ragonese@studenti.unipg.it (F.R.); lorenzo.monarca@unipg.it (L.M.); federica.bastioli@unipg.it (F.B.); loretta.mancinelli@unipg.it (L.M.); 3Department of Experimental Medicine, Perugia Medical School, University of Perugia, Piazza Lucio Severi 1, 06132 Perugia, Italy; cataldo.arcuri@unipg.it; 4S&R Farmaceutici S.p.A Bastia Umbra, 08063 Perugia Italy; r.iannitti@srfarmaceutici.com; 5Department of Biology, Faculty of Natural Science and Health Care, Batumi Shota Rustaveli State University, 6010 Batumi, Georgia; irina.nakashidze@bsu.edu.ge; 6Department of Veterinary Science, University of Perugia, Via San Costanzo 4, 06126 Perugia, Italy; gabriele.brecchia@unipg.it (G.B.); laura.menchetti7@gmail.com (L.M.); 7Department of Pharmaceutical Sciences, University of Perugia, Via A. Fabretti 48, 06123 Perugia, Italy; michela.codini@unipg.it

**Keywords:** resveratrol, magnesium dihydroxide, solubility, bioavailability, dissolution rate, microparticles

## Abstract

Resveratrol, because of its low solubility in water and its high membrane permeability, is collocated in the second class of the biopharmaceutical classification system, with limited bioavailability due to its dissolution rate. Solid dispersion of resveratrol supported on Magnesium DiHydroxide (Resv@MDH) was evaluated to improve solubility and increase bioavailability of resveratrol. Fluorimetric microscopy analysis displays three types of microparticles with similar size: Type 1 that emitted preferably fluorescence at 445 nm with bandwidth of 50 nm, type 2 that emitted preferably fluorescence at 605 nm with bandwidth of 70 nm and type 3 that is non-fluorescent. Micronized pure resveratrol displays only microparticles type 1 whereas type 3 are associated to pure magnesium dihydroxide. Dissolution test in simulated gastric environment resveratrol derived from Resv@MDH in comparison to resveratrol alone displayed better solubility. A 3-fold increase of resveratrol bioavailability was observed after oral administration of 50 mg/kg of resveratrol from Resv@MDH in rabbits. We hypothesize that type 2 microparticles represent magnesium dihydroxide microparticles with a resveratrol shell and that they are responsible for the improved resveratrol solubility and bioavailability of Resv@MDH.

## 1. Introduction

Resveratrol (trans-3,5,4′-tri-hydroxic-stilbene) is a stilbenic structure polyphenol, initially isolated from the root of the white hellebore (*Veratrum Grandiflorum O. Loes*) and later from the root of the *Polygonum cuspidatum*, a plant used in traditional Chinese and Japanese medicine. Resveratrol became popular in 1992 when it was suggested that it could be the reason behind red wine’s cardio-protective effects (French paradox; [1]), and its popularity increased in 1997 when it was proven that resveratrol was able to prevent colorectal cancer in mice [1]. Resveratrol based compounds present anti-oxidant, anti-inflammatory, anti-viral, cardio-protective, neuro-protective, anti-cancer and anti-angiogenetic activities [1,2,3]. It has been recently observed in obese human subjects that treatment with trans-resveratrol reduces glucose, triglycerides and inflammatory marker levels with a similar effect to the one induced by caloric restriction [4]. The mechanism of action of resveratrol has not been completely defined yet, and for this reason recently studies have been carried out in order to understand the aspects that are still not clear [4].

Resveratrol is poorly bioavailable because of reduced absorption mainly due to its low solubility and fast metabolism that converts it into glucuronide and sulfates compounds [1,5]. In humans resveratrol can be detected in plasma about 30 min after oral administration, meaning that its absorption already starts at the gastric level and reaches a plasmatic submicromolar concentration peak. Such peak is variable and hardly related to the used dose. For example, by administrating a 25 mg dose of resveratrol a 10 ng/mL plasmatic concentration is obtained, while increasing such dose by 20 times (500 mg/day) its plasma level increases only seven times (72.6 ng/mL) [6]. Differences in resveratrol absorption have been demonstrated by clinical trials based on the oral administration of 150 mg/day of resveratrol for a prolonged period of time. It has been observed that the same dose produces different plasmatic concentrations: 231 ng/mL [4] and 24.8 ng/mL [7]. Several strategies have been performed to increase its bioavailability and improve its potential health properties. A recent revision of the literature highlights how the increased bioavailability of resveratrol is a necessary element in order to evaluate the real pharmaceutical and health potential of this well-known polyphenol [5]. According to the biopharmaceutical classification system (BCS) [8,9], resveratrol belongs to the second class which means that it is characterized by low solubility in water (about 30 mg/L), while it shares a high membrane permeability (log P~3.1) [10]. Among the different strategies, new formulations have been developed that are able to increase its apparent solubility for example by using a lipophilic vehicle or through various processes such as the complexation with cyclodextrins, nanopreparation, or micellar solubilization with biliary acid [10,11,12]. It has been demonstrated, in in vitro studies, that the increase of apparent resveratrol solubility allows a partial saturation of the mechanisms that are involved in its metabolism (conjugation) with a subsequent increase of resveratrol’s bioavailability [13]. This is in accordance with BCS for molecules class II that increasing resveratrol apparent solubility produces a bioavailability improvement [8,9,14], but in a dedicated study the increased solubility with cyclodextrins doesn’t modify its bioavailability [12].

In the present study we investigated that the solid dispersion of resveratrol on magnesium dihydroxide increases its solubility and bioavailability indicating that in some instance this approach could be exploited to enhance biological properties of resveratrol. Although resveratrol does not display chelating properties, some studies have shown its ability to interact with heavy metals such as copper, zinc and aluminum [15,16]. In this work we report that resveratrol interacts with magnesium dihydroxide at the microparticle level and that this is able to modify its bioavailability.

## 2. Material and Methods

### 2.1. Solid Dispersion of Resveratrol on Magnesium Dihydroxide Preparation

Magnesium dihydroxide and resveratrol (from *Polygomun cuspidatum,* 98% pure) solid dispersion was performed by modified co-precipitation method of Biswicka et al. [17]. Magnesium dihydroxide on resveratrol solid dispersion and pure micronized resveratrol described in this study was obtained by Good Manufacturing Practice (GMP) chain by Prolabin & Tefarm, Ponte Felcino (PG) and distributed by S&R Farmaceutici SpA, Via dei Pioppi 2, 06083 Bastia Umbra (PG), with the trade name, Revifast^®^(produced by the manufacturer La Sorgente del Benessere, Via Prenestina, 141 -02014 Fiuggi (FR) Italy on behalf of S&R Farmaceutici S.p.A) The resveratrol content in the solid dispersion was evaluated using the HPLC method (see below). The mean value obtained in the three samples was about 30% and 70% of total weight of resveratrol and magnesium dihydroxide, respectively.

### 2.2. Particle Size Analysis

The size of the particles was determined using a Malvern Mastersizer 2000, a laser diffraction particle size analyzer, for the dried powders.

### 2.3. Dissolution Assays

A weighed amount of RSV@MDH or resveratrol were placed in series of closed flat-bottomed glass vessels containing 250 mL of Simulated Gastric Fluid (SGF). The composition of SGF was 35 mM of NaCl, pH 1.2 with HCl. The vessels were inserted in shaking water bath (Nuve ST 30) at 37 °C and 110 rpm for 2 h. At appropriate times (1, 3, 5, 10, 15, 20, 30, 45, 60, 90 and 120 min) 2 mL samples were withdrawn and replaced by fresh dissolution medium, then filtered (Spartan 13/02 RC, Whatman GmbH, Dassel, Germany) and analyzed. The drug concentration was determined by HPLC (see below).

### 2.4. Field Emission Scanning Electron Microscopy

The morphology of the samples was investigated with a FEG LEO 1525 scanning electron microscope (FE-SEM). FE-SEM micrographs were collected by depositing the samples on a stub holder and after sputter coating with chromium for 20 s.

### 2.5. Fluorescence Microscopy

Microscopic fluorescence analysis of powders was performed using an Axio Esaminer (Zeiss, Jena, Germany) fluorescence microscope with a CCD digital camera Axio Cam 502 Mono. Samples have been observed with DAPI filter (G 365, FT 395, BP 445/50), and with Rhodamine (BP 545/25, FT 570, BP 605/70) using for excitation mercury lamp (HXP 120V). Image acquisition and analysis was performed with Zen 2 software (Zeiss, Jena, Germany).

### 2.6. In Vivo Absorption Test

The trial was carried out at the experimental farm of the University of Batumi, Georgia. Rabbits were exposed to a continuous photoperiod of 16 h light per day at 40 lx. Room temperature ranged from 18 to 27 °C. Fresh water was always available. Animals were fed with 130 g/day of a standard diet. The experimental protocol was approved by the Local Ethical Committee for Animal Experimentation at the University Batumi, Georgia. All efforts were made to minimize animal distress and to use only the number of animals necessary to produce reliable results. The tests were conducted on New Zealand White hybrid rabbits (4.5–5 kg weight range). Two groups of four animals each were prepared for the comparative treatment of resveratrol (pure resveratrol versus Resv@MDH). The rabbits were fasted for 24 h before administration of a suspension containing 50 mg/kg of pure resveratrol or 50 mg/kg of resveratrol from Resv@MDH according to Jaisamut et al., 2017 [18]. The powders were suspended in 10 mL of a glucose solution and orally administered to a conscious animal by a syringe (0 min). At 0, 5, 15, 30, 45, 90, 120 and 180 min, blood samples were taken (about 2 mL) through the auricular artery and put in heparinized tubes. The samples were centrifuged at 2500 g for 5 min and the plasma was recovered. Acetonitrile was added to the plasma samples (*v*:*v* 1:1 ratio) and left for 5 min in order to precipitate plasma proteins. After centrifugation the supernatant was recovered for the dosage of resveratrol by HPLC.

### 2.7. HPLC Analysis

The measurements were performed by an Agilent HPLC 1200 series equipped with an Agilent Zorbax SB C18 4.6 × 250 mm 5-μm Agilent P/N 880975-902 column. Elution was carried out under isocratic conditions using as mobile phase (Water + 0.1% *v*/*v* Trifluoroacetic acid)/(Acetonitrile + 0.1% *v*/*v* Trifluoroacetic acid) = 65/35, with a flow of 1mL/min and a column temperature of 30 °C. A total of 20 μL of samples were injected, after 0.2 μm Nylon membrane filtration, and the analytes were detected by VWD Detector, *λ* = 306 nm. For the quantification of resveratrol a calibration was performed to detect the polyphenol at a retention time of 5.6 min with a detection limit of 4 ng/mL. All the plasma concentrations were multiplied by 2 to take into account the dilution in acetonitrile during sample preparation and by 3.6 to take into account the yield of extraction of resveratrol from plasma (28%) [19].

### 2.8. Statistical Analysis

All results are expressed as the mean ± SE. Differences between two related parameters were assessed by Student’s *t*-test. Differences were considered significant at *p* < 0.05. The number of animals used in the current experimental trial is based on the work by Jaisamut et al., 2017 [18].

## 3. Results

### 3.1. Microscopic Analysis of Solid Dispersion of Resveratrol on Magnesium Dihydroxide

RSV@MDH powder was dispersed in glycerol and was observed by bright-field microscopy. The presence of particles with different scattering profiles in a narrow size range of a few micrometers was observed (Figure 1A). Fluorescence analysis of the samples with DAPI filter showed that about 10–20% of the microparticles emitted fluorescence. These microparticles were defined as type 1 (Figure 1B). The mean size of type 1 microparticles was 1.8 ± 0.1 μm, *n* = 40 in diameter (given the non-spherical morphology of the particles, the longest diameter has been taken into account). When the sample was analyzed with the rhodamine filter, a comparable population of particles was visualized with a mean size of 2.0 ± 0.2 μm, *n* = 34 and was named type 2 (Figure 1C). Type 1 microparticles displayed very scant signals when observed with the rhodamine filter similar to the type 2 particles with the DAPI filter. Finally, the majority of the microparticles didn’t display any fluorescence in either filter and were defined as type 3 and had medium size similar to others (Figure 1D). 

Thus, the solid dispersion of resveratrol on magnesium dihydroxide was composed by three distinct populations of microparticles based on the fluorescence profile. When we similarly analyzed the dry powder without dispersion in glycerol we observed aggregates of size around 5 μm were present as a possible consequence of the interaction of the three types of microparticles (Figure 2A). In accordance, that the aggregates are based on different types of microparticles, they displayed fluorescence signals from every channel. Granulometric and SEM analysis showed two distinct population sizes, one with size around 1 µm and the second population with size around 6 µm of diameter (Figure 2B,C).

### 3.2. Molecular Nature of Microparticles of Solid Dispersion of Resveratrol on Magnesium Dihydroxide

To define the molecular nature of the different types of microparticles, we studied a powder of pure micronized resveratrol with similar distribution size of solid dispersion. Granulometric analysis confirmed that micronized resveratrol have the size of 1–6 µm in diameter (Figure 3A) similar to the particles size of RSV@MDH (see Figure 1 for comparison). Fluorescence microscopy analysis of the micronized resveratrol displayed all the microparticles emitted fluorescence intensity as type 1 particles (Figure 3B,C), whereas the presence of microparticles that showed fluorescent properties as type 2 and 3 were not observed (Figure 3D,E). No fluorescence was observed (DAPI and Rhodamine filters) during microscopic analysis of pure magnesium dihydroxide, indicating that type 3 microparticles could be constituted by only magnesium dihydroxide. These data suggested that the type 1 microparticles were microparticles of pure resveratrol, whereas the type 3 microparticles represented magnesium dihydroxide. Since the fluorescence properties were due to resveratrol, type 2 macroparticles could be distinguished from type 3 microparticles by presence of resveratrol. The type 2 microparticles were further investigated to define the morphological features. In fact, it was possible to see a shell of fluorescence around a non-fluorescent core and this was due to resveratrol surrounding the core of magnesium hydroxide microparticles (Appendix A). All the features of the microparticles are stated in Table 1.

### 3.3. Dissolution of Solid Dispersion of Resveratrol on Magnesium Dihydroxide. 

In Figure 4 dissolution profiles of Resv@MDH (red squares) and pure resveratrol (black squares) are presented (mg/L in function of time). The experimental data was fit with exponential equation C(t) = Cmax (1 − exp(−t/τ)), where Cmax = maximum solubility value; t = time; τ = time in which dissolution reaches about 63% of maximum process. The equation represents a form studying the dissolution profiles according to Weibull’s models [20]. The best data fit is for Cmax: 40.8 and 13 mg/L for Resv@MDH and resveratrol respectively while τ was 0.4 and 2.2 min for Resv@MDH and resveratrol respectively. These data indicated that Resv@MDH showed a dissolution rate five times higher than resveratrol (compared to τ) and a maximum solubility three times as big (compared to Cmax). To assess the importance of particles size in dissolution rate, we compared the solubility profile of pure micronized resveratrol with similar size particles of Resv@MDH (Figure 3 and Figure 4). It was possible to see (compare black and green squares in Figure 4A) the reduction of particles size modified only dissolution kinetic according to the Noise-Witting law, but did not modify the maximal solubility [21]. 

To verify if magnesium ion participates in major solubility (Cmax) of resveratrol by forming a complex, we verified the interaction between them by performing spectrophotometric profile of resveratrol alone or in presence of magnesium ion in acid environment. It is possible to see in Figure 4B, that the addition of magnesium does not significantly modify the UV absorption spectra, suggesting that the magnesium does not interact with resveratrol and that the major solubility was dependent on other factors.

### 3.4. Pharmacokinetic Profile of Solid Dispersion of Resveratrol on Magnesium Dihydroxide

The rabbit animal model is excellent to perform pharmacokinetic studies [20] and recently was used to evaluate the bioavailability of a new resveratrol formulation [18]. The mean plasma concentration of resveratrol following oral administration of 50 mg/kg of Resv@MDH and pure resveratrol was investigated in the rabbit animal model. Resveratrol plasma concentration versus time curves from administration is displayed in Figure 5. Pharmacokinetic variables derived from this pharmacokinetic profile are summarized in Table 2. Resveratrol is virtually absent in animal plasma prior to oral administration (0 min) but it seemed to be rapidly absorbed with a peak of maximal concentrations (C_max_) between 15 and 30 min post-dose. The C_max_ of resveratrol was 76.3 ng/mL and 101.3 ng/mL for resveratrol and Resv@MDH respectively. At 30 up to 90 min from the administration, the resveratrol plasma concentration of Resv@MDH treated animals results statistically greater as compared to resveratrol treated animal, while at 180 min the resveratrol is no longer detectable in the plasma of both groups of animals. The values of Area Under Curve (AUC) of the plasma concentration profile until the 3-h time point was 2698 ng min/mL and 8944 ng min/mL or resveratrol and Resv@MDH respectively. This data demonstrates an enhancement of resveratrols bioaviability by 3.3-fold (ratio of AUC_Resv@MDH_/AUC_resveratrol,_
Table 2).

## 4. Discussion

Solid dispersion of resveratrol on magnesium dihydroxide (Resv@MDH) represents a new formulation that possesses an increased solubility of resveratrol (spring form). Resv@MDH is able to solubilize itself faster and in greater amounts with respect to resveratrol, with remarkable advantages in biopharmaceutical terms and therefore of bioavailability. From the physical point of view it is a polydisperse granular material, where the active is supported by inorganic material with a high safety level (magnesium hydroxide). Furthermore, this improves its performance without chemically modifying the natural product’s structure. Resv@MDH allows to obtain an apparent solubility much higher with respect to resveratrol as a consequence of an increased dissolution rate and of the establishment of over-saturation phenomena due to different energetic states of resveratrol (Figure 6). This dispersion is formed by three types of microparticles that we define as type 1, 2 and 3. Based on our results we hypothesized that microparticles type 1 and 3 represent resveratrol and magnesium dihydroxide crystals, respectively. The unexpected result is the presence of microparticles type 2 that probably represent the form responsible for enhanced properties of the solid dispersion. Based on the evidence of change of its fluorescence, we suggest that a fraction of resveratrol forms a shell around magnesium dihydroxide microparticles. The better solubility of resveratrol displayed by solid dispersion could be explained by the coexistence of two energetic states of resveratrol related to the two types of microparticles observed (type 1 and 2, Figure 6). It is possible to exclude the involvement of free magnesium ions (Mg^2+^) in improving the solubility of resveratrol since their absorbance spectrum was not modified by the presence of metals in acidic environment (Figure 4B). The state of over-saturation could lead to the major absorption (increase of the gradient concentration) and therefore increased bioavailability [22].

The reduced and homogeneous particle size represents a parameter that improves the dissolution rate observed for Resv@MDH is according to Noyes and Whitney law [21]. The comparative dissolution rate of resveratrol displayed in Figure 5 demonstrates that the supersaturating state is not dependent from the particles size of resveratrol. Further studies are needed to clarify the mechanisms of the better solubility of solid dispersion of resveratrol on magnesium dihydroxide. The Resv@MDH represent a new way to uncover the therapeutic potential of resveratrol with possible application as anti-inflammatory, anti-viral, cardio-protective, neuro-protective, anti-cancer and anti-angiogenetic agent [1,2,3]. As regard the anticancer properties, resveratrol was demonstrated to increase the effect of radio and chemotherapeutic agents [23] in particular against glioblastoma cancer cells [24].

## Figures and Tables

**Figure 1 nutrients-10-01925-f001:**
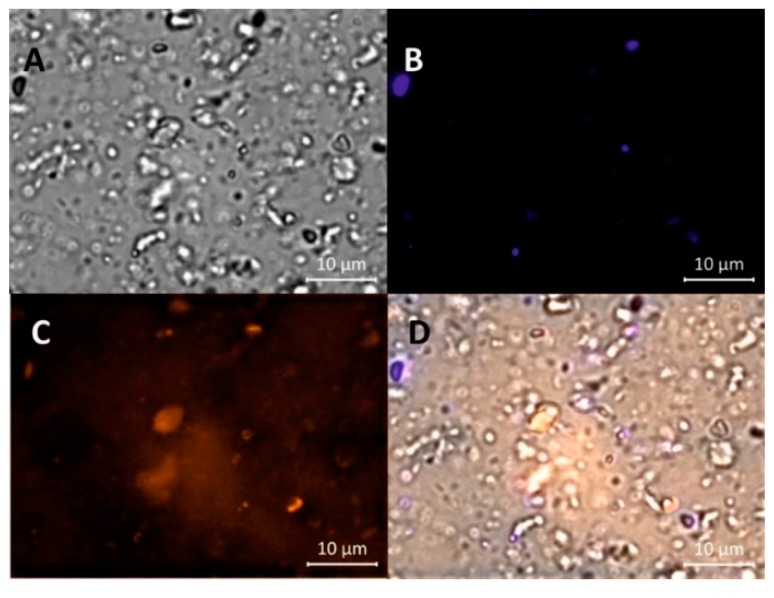
Image of Resv@MDH powder dispersed in glycerol under different excitation sources. (**A**) Bright-field; (**B**) DAPI (4′,6-diamidino-2-phenylindole) fluorescence filter; (**C**) Rhodamine fluorescence filter; (**D**) merging of the Bright-field, DAPI and Rhodamine images.

**Figure 2 nutrients-10-01925-f002:**
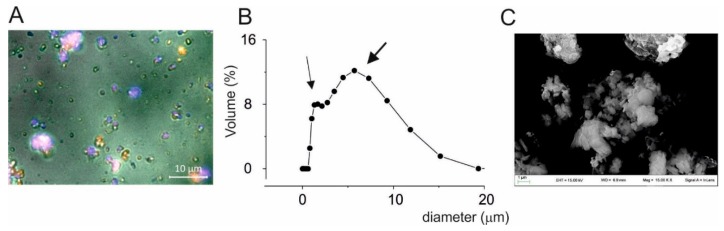
Properties of Resv@MDH dry powder. (**A**) Image created by digital merging of bright-field and DAPI/Rhodamine fluorescence illumination. (**B**) Granulometric analysis of Resv@MDH dry powder. (**C**) SEM image of Resv@MDH dry powder.

**Figure 3 nutrients-10-01925-f003:**
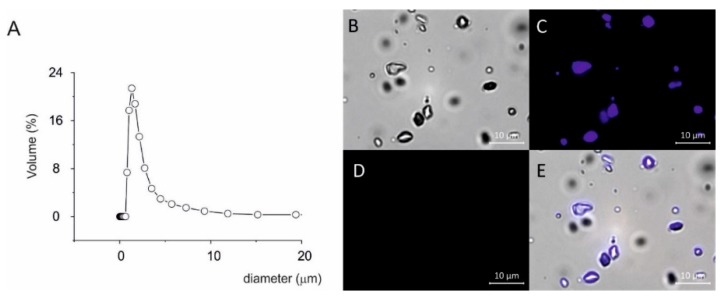
Properties of pure micronized resveratrol. (**A**) Granulometric analysis of microcrystalline resveratrol. (**B**–**E**) Image of crystalline resveratrol powder dispersed in glycerol under different excitation sources. (**B**) Bright-field; (**C**) DAPI fluorescence filter; (**D**) Rhodamine fluorescence filter. (**E**) Merging of the Bright-field, DAPI and Rhodamine images.

**Figure 4 nutrients-10-01925-f004:**
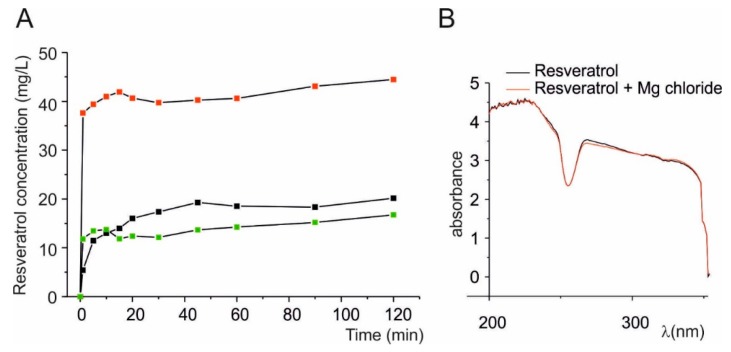
Solubility of resveratrol from Resv@MDH and its interaction with magnesium ion. (**A**) dissolution test of pure resveratrol powder (black MDH square) versus solid dispersion on magnesium dihydroxide (Resv@MDH, red square) and pure micronized resveratrol (green square). (**B**) UV/Vis Absorbance spectroscopy for the study of Resveratrol. Black: 0.008 mM Resveratrol in ethanol:water (75:25, *v*/*v*) in 100 mM HCl; Red: 0.008 mM Resveratrol in ethanol:water (75:25, *v*/*v*) in 100 mM HCl + 0.008 mM of MgCl.

**Figure 5 nutrients-10-01925-f005:**
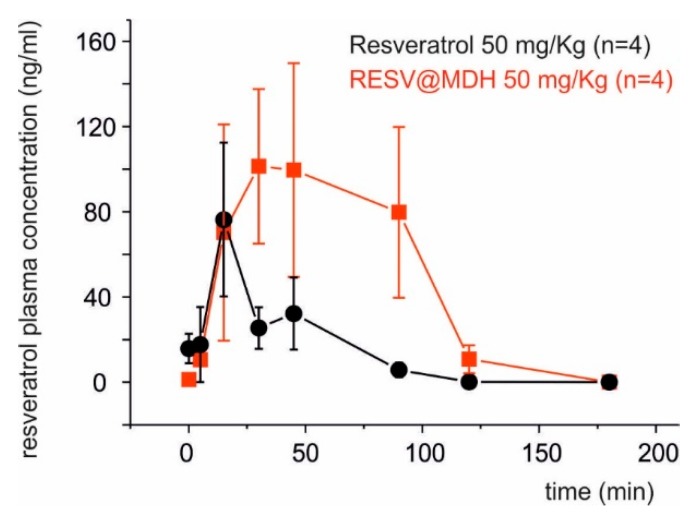
Pharmacokinetic profiles of resveratrol after oral administration in rabbits. Groups of 4 animals each were treated with resveratrol (50 mg/Kg of pure resveratrol versus Resv@MDH). Blood samples taken at 0, 5, 15, 30, 45, 90, 120 and 180 min.

**Figure 6 nutrients-10-01925-f006:**
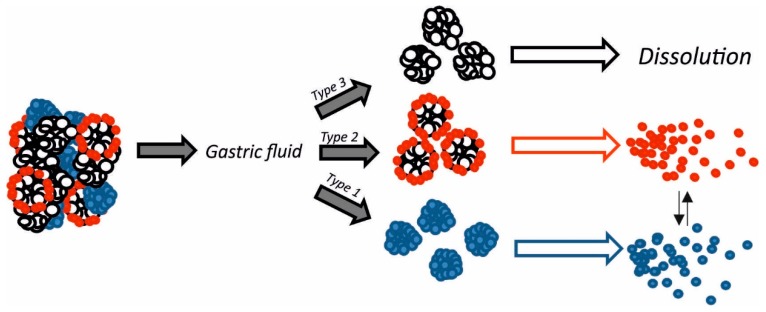
Scheme of hypothetical dissolution events that occur to Resv@MDH powder when it is in contact with simulated stomach fluids. Big powder aggregates divide into three main microparticles named as type 1, 2 and 3. In acidic milieu the type 3 microparticles of magnesium dihydroxide completely dissolves; type 1 microparticles (blue) dissolves in water solution together with type 2 microparticles (red). In this case the limiting step in resveratrol release could be related to acid erosion of dihydroxide core.

**Table 1 nutrients-10-01925-t001:** Principle features of microparticles of RSV@MDH.

Characteristic	Type 1 Microparticles	Type 2 Microparticles	Type 3 Microparticles
DAPI filter (G 365, FT 395, BP 445/50)	High intensity	Low intensity	none
Rhodamine (BP 545/25, FT 570, BP 605/70)	Low intensity	High intensity	none
Particles size	~1.8 ± 0.1 μm	~2.0 ± 0.2 μm	~1.7 ± 0.1 μm
Resveratrol contents	High	Low (shell distribution)	none
Dissolution rate	Low	High	n.d.

**Table 2 nutrients-10-01925-t002:** Pharmacokinetic parameters of oral administration of 50 mg/Kg of resveratrol from pure resveratrol and from Resv@MDH.

Parameters	Resveratrolo 50 mg/Kg	Resv@MDH (Resveratrol 50 mg/Kg)	Increase %
AUC (Area Under Curve)	2698 ng min/mL	8944 ng min/mL	330
Time to plasmatic peak	15 min	30 min	200
Peak duration	25 min	105 min	420
Cmax	76.3 ng/mL	101.3 ng/mL	130

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
