# Peer review of "Solid Dispersion of Resveratrol Supported on Magnesium DiHydroxide (Resv@MDH) Microparticles Improves Oral Bioavailability"

_nutrients, 2018, doi:10.3390/nu10121925_

Reviewer 1 Report

General comments

The work deals with a solid dispersion of resveratrol with magnesium dihydroxide marketed with the tradename Revifast (RESV@MDH).

The method of preparation has been already used for the production of Revifast and the authors attempt to explain the in-vitro and in-vivo improvement using fluorimetric microcsopy.

Although the rapid initial release suggests presence of drug on the surface of microparticles, the evidence based on the assumption that the change of fluorescence, arises from a fraction of resveratrol forming a shell around magnesium dihydroxide microparticles is not adequate and further confirmatory information should be provided.

Furthermore, the part of ‘granulometry’ in the work does not seem to add any information for the different in-vitro and in-vivo performance, and the presentation of two size analysis, one for dispersed and a second for aggregated particles complicates the issue further.

More importantly, since any type of fluorescence behavior is possible from Mg dihydroxide particles, how did the authors came to the conclusion that fluorimetry differentiates between these and type 2 particles?

Specific points

Line 24. Clear differences in granulometric analysis is not obvious from the results of the work.

Line 30. “According to the biopharmaceutical classification” Delete.

Line 32 Could you separate type 2 microparticles from the rest?

Lines 48-56. How relevant is this text to the purpose of the work?

Line 85. Remove “present”.

Lines 98-100. The way authors presented the method in the beginning of the paragraph is as if the solid dispersion was prepared by them. However, since the Magnesium dihydroxide on resveratrol solid dispersion was obtained from outside this should be made clear in the beginning of the paragraph.

Line 106. Why ‘Supersaturation assays’ and not simply ‘Solubility measurement’?

Lines 121, 122. Text does not make sense.

Line 134. Remove ‘oral’.

Lines 189-192. Since ‘any fluorescence can be observed (DAPI and Rhodamine filters) during microscopic analysis of pure magnesium dihydroxide’ how does this method differentiate them from type 2?

Figure 4. Please designate the curves according to the colours.

Lines 277-283. Author contributions section has not been completed.

Reviewer 2 Report

Please describe how the 3R rules were applied in the 2.6 section.

Is there any dose caution with the ingestion of Mg? How was the Mg dose that the animals ingested?

Why did you select the 50 mg/kg dose and no other dose?

Did you perform subsequent analysis of resveratrol excretion metabolism? These data would enhance the quality of the manuscript significantly.

Your hypothesis related with the dissolution events could be tested with in vitro digestion protocols. Do you aim to perform such study?

Several typo mistakes are in the manuscript (lack of space between two or more words, "resveratrolo" in Table 1, sovrasaturation is oversaturation). Author contributions is not completed.

Author Response

Please describe how the 3R rules were applied in the 2.6 section.

a. Replacement

The in vitro experimental model does not allow the study of the biodistribution of a molecule after oral administration, thus making the in vivo experimental model is irreplaceable.

All compounds administered orally undergo a complex phenomena of digestion and absorption that can be evaluated only with an in vivo model.

For this reason,the rabbit was chosen as the preferred animal model because, in relation to the objectives and the procedures (administration of the substance, blood collection, etc) of the research project, it resulted as better suited to be used. Moreover, some authors have a great experience with this species (Menchetti et al., 2018; Collodel et al., 2015; Brecchia et al., 2014).

b. Reduction

In this project we reduced to a minimum the number of animals used. The number of the animals used in the current experimental trial was based on the work by Jaisamut et al., 2017.

c. Refinement

The present group has a direct experience and expertise in this species of interest (rabbit): therefor, we adopted all their practice in order to minimize animal suffering.

In particular, the animals had very minor manipulation based on the administration of the resvetrarol and the following blood collections. To reduce the pain derived by the puncture of the vessels we used a cream, applied locally containing an anesthetic such as lidocaine and prilocaine (EMLA, Aspen Pharma Schweiz GmbH, Baar). We were also prepared, if any case required the animal to be sedated, in order to minimize handling stress, through the use of medetomidine (0.2 mg / kg) and ketamine (10 mg / kg), but in our case this was not necessary throughout the procedure during the experimental trial. The trial was carried out under the direct supervision of a veterinary, who assessed the state of health of the animals and the correctness of all the procedures. The molecule that was administered, at the intended dose and did not dot show any toxicity or harmful side effects.

References

1) Menchetti L, Barbato O, Filipescu IE, Traina G, Leonardi L, Polisca A, Troisi A, Guelfi G, Piro F, Brecchia G. Effects of local lipopolysaccharide administration on the expression of Toll-like receptor 4 and pro-inflammatory cytokines in uterus and oviduct of rabbit does. Theriogenology. 2018 Feb;107:162-174.

2) Collodel G, Moretti E, Brecchia G, Kuželová L, Arruda J, Mourvaki E, Castellini C. Cytokines release and oxidative status in semen samples from rabbits treated with bacterial lipopolysaccharide. Theriogenology. 2015 Apr 15;83(7):1233-40.

3) Brecchia G, Menchetti L, Cardinali R, Castellini C, Polisca A, Zerani M, Maranesi M, Boiti C. Effects of a bacterial lipopolysaccharide on the reproductive functions of rabbit does. Anim Reprod Sci. 2014 Jun 30;147(3-4):128-34.

4) Jaisamut P, Wiwattanawongsa K, Wiwattanapatapee R. A Novel Self-Microemulsifying System for the Simultaneous Delivery and Enhanced Oral Absorption of Curcumin and Resveratrol. Planta Med. 2017 Mar;83(5):461-467. doi: 10.1055/s-0042-108734. Epub 2016 Jun 9. PubMed PMID: 27280934.

Furthermore, we included a new paragraph in material and methods section at  the line 251.

2.8 Statistical analysis

All results are expressed as the mean ± se. Differences between two related parameters were assessed by Studentʼs t-test . Differences were considered significant at p < 0.05. The number of the animals used in the current experimental trial is based on the work by Jaisamut et al., 2017.

Is there any dose caution with the ingestion of Mg? How was the Mg dose that the animals ingested?

Since Magnesium hydroxide in RESV@MDH represents the 70%, the Mg in RESV@MDH represent about 30% (about 50 mg/Kg). Thus, the Mg dose that the animals ingested was about 50 mg/Kg. In human hypermagnesemia is rare and occurs occasionally in patients receiving Mg2+-containing parenteral infusions, especially in patients with renal failure, and it is associated with mild symptoms such as nausea/vomiting, bradycardia, and flushing to flaccid paralysis of voluntary muscles, complete heart block, and coma (Jertiger et al., 2015). None of the above reported symptoms were observed following oral administration of RESV@MDH in our study. These date indicated that the dose of about 50 mg/Kg is safe.

Jertiger G, Jones E, Dahdal DN, Marshall DC, Joseph RE. Serum magnesium concentrations in patients receiving sodium picosulfate and magnesium citrate bowel preparation: an assessment of renal function and electrocardiographic conduction. Clin Exp Gastroenterol. 2015 Jul 28;8:215-24. doi:10.2147/CEG.S79216. eCollection 2015.

Why did you select the 50 mg/kg dose and no other dose?

The dose of 50 mg/kg was chosen according to previously reported pharmacokinetic study of resveratrol and that used the same rabbit animal model (Jaisamut et al., 2017).

We will add in the result section at the line 424 the follow sentence “The rabbit animal model is excellent to perform pharmacokinetic studies and recently was used to evaluate the bioavailability of a new resveratrol formulation (Menchetti et al., 2000, Jaisamut et., 2017)”.

We will added in the material and methods section at the paragraph 2.6 at the line 252 the follow sentence "…according to Jaisamut et al., 2017”.

Menchetti L, Canali C, Castellini C, Boiti C, Brecchia G. The different effects of linseed and fish oil supplemented diets on insulin sensitivity of rabbit does during pregnancy. Res Vet Sci. 2018 Jun;118:126-133; Footea R.H., Edward W. Carneyb E.W. The rabbit as a model for reproductive and developmental toxicity studies. Reproductive Toxicology 14 (2000) 477–493.

Jaisamut P, Wiwattanawongsa K, Wiwattanapatapee R. A Novel Self-Microemulsifying System for the Simultaneous Delivery and Enhanced Oral Absorption of Curcumin and Resveratrol. Planta Med. 2017 Mar;83(5):461-467. doi: 10.1055/s-0042-108734. Epub 2016 Jun 9. PubMed PMID: 27280934.

Did you perform subsequent analysis of resveratrol excretion metabolism? These data would enhance the quality of the manuscript significantly.

Thank you for the suggestion. At the moment our study mainly addressed the absorption process of resveratrol, where the interaction with magnesium di hydroxide has been expected to be relevant. Thus, for this reason we have not yet investigated further resveratrol excretion metabolism.  

Your hypothesis related with the dissolution events could be tested with in vitro digestion protocols. Do you aim to perform such study?

Our dissolution test was performed with gastric simulated fluid that mimics a digestion environment in in vitro. To better describe this method we specified the composition of gastric simulated fluid by adding the following sentence in the methods section: …. Simulated Gastric Fluid (SGF). The composition of SGF was 35 mM of NaCl, pH 1.2 with HCl.

Several typo mistakes are in the manuscript (lack of space between two or more words, "resveratrolo" in Table 1, sovrasaturation is oversaturation). Author contributions is not completed.

We will correct the text according to your observation and complete Author contribution according to your observation.

Round  2

Reviewer 1 Report

I advise the authors to do some language editing an clarification of discussion points in order to present a clearer, neater submission. 

Reviewer 2 Report

The manuscript has been conveniently reviewed according to the suggestions. I recommend to publish it.